# Exploring the Conformation and Thermal Stability of Human Serum Albumin Corona of Ferrihydrite Nanoparticles

**DOI:** 10.3390/ijms21249734

**Published:** 2020-12-20

**Authors:** Claudia G. Chilom, Adriana Bălan, Nicoleta Sandu, Maria Bălăşoiu, Sergey Stolyar, Oleg Orelovich

**Affiliations:** 1Faculty of Physics, University of Bucharest, Str Atomiștilor 405, CP MG 11, RO-077125 Bucharest, Romania; andronie@3nanosae.org (A.B.); s.nicoleta59@yahoo.ro (N.S.); 2Joint Institute for Nuclear Research, Joliot-Curie No.6, 141980 Dubna, Russia; masha.balasoiu@gmail.com (M.B.); orel@jinr.ru (O.O.); 3Moscow Institute of Physics and Technology, Institutskiy Per. No. 9, 141701 Dolgoprudniy, Russia; 4Horia Hulubei National Institute of Physics and Nuclear Engineering, RO-077125 Măgurele, Romania; 5Krasnoyarsk Scientific Center, Federal Research Center KSC SB RAS, Akademgorodok St. No. 50, 660036 Krasnoyarsk, Russia; stol@iph.krasn.ru

**Keywords:** ferrihydrite nanoparticles, human serum albumin, protein stability, molecular docking

## Abstract

In the last few years, a great amount of attention has been given to nanoparticles research due to their physicochemical properties that allow their use in analytical instruments or in promising imaging applications on biological systems. The use of ferrihydrite nanoparticles (Fh-NPs) in practical applications implies a particular control of their magnetic properties, stability, biocompatibility, interaction with the surface of the target, and low toxicity. In this study, the formation and organization of human serum albumin (HSA) molecules around the simple Fh-NPs and Fh-NPs doped with Co and Cu were examined by Scanning Electron Microscopy (SEM) and Atomic Force Microscopy (AFM) in terms of morphology and particle size. The topology of all Fh-NPs shows an organized area of HSA around each type of Fh-NP. Molecular docking studies were used in order to determine the probable location of the ferrihydrite in the HSA structure. The thermal stability of these nanohybrids was further investigated by fluorimetry, using 214-Trp residue from HSA as a spectral sensor. The denaturation temperature (*T_m_*) was determined, and stabilization of the HSA structure in the presence of Fh-NPs was discussed. This study could be a starting point for the development of different applications targeting the structure and stability of Fh-NPs complexes with proteins.

## 1. Introduction

The research of nanoparticles (NPs) deserves a great amount of attention due to the physicochemical properties that allow their use in analytical instruments or in promising imaging applications on biological systems [1]. Once entered into a biological fluid, NPs interact with different proteins, forming around them, the so-named protein corona [2,3], which may be of great interest in therapeutic applications. The identity, composition, and concentration of proteins [4], the function and the properties of NPs (e.g., shape, net charge, surface coating) [4,5], together with the environmental factors (as pH and temperature) influence the ability of the NPs to adsorb proteins. The architecture of the protein corona is unique, and it can influence the biological effects of NPs. For example, coating the NPs with corona proteins can reduce the ability of NPs to reach their cellular target [6].

The interactions of NPs with proteins can be approached by means of several analytical methods and strategies with the purpose of obtaining information related to the interaction strength, stoichiometry, and conformation changes induced by NPs–protein binding and also, the kinetics [7].

A widespread way to transport ligands to/into the cells is their binding to serum albumins, which is followed by the movement of the complex to the cell target. It is known that bovine (BSA) and human serum albumin (HSA) bind and carry, in a manner depending on temperature and pH [8], a very large variety of small molecules (i.e., ligands), from marker molecules [9] to anticancer drugs—such as mitomycin [10] or tetracaine [11], hormones [12], bioactive plant elements as folic acid [13,14,15], but also to different types of NPs—such as iron oxide nanoparticles [16].

Hydrated iron (III) oxide is commonly found in the form of NPs, called ferrihydrite (Fh). Ferrihydrite nanoparticles (Fh-NPs), with and without magnetic properties [17,18], are designed for drug delivery [19] or medical diagnosis and therapy [20]. In the recent years, due to the fact that iron nanoparticles seems to be not toxic, their use in different fields was steadily increased [21,22]. The use of Fh-NPs in practical applications implies a particular control of magnetic properties, stability, biocompatibility, interaction with the surface of the target, and also of their toxicity [23]. Fh-NPs properties can be improved by building functional structures together with biomolecules [24], polymers, or metals [25,26]. Knowledge of HSA corona particularities can be useful to determine the use of Fh-NPs in the medical field [27].

HSA and other proteins adsorption onto NPs surface are the first line of defense against foreign agents [28], aiming to neutralize and eliminate these invading agents. Therefore, it is important to know how HSA structure and properties are modified after its adsorbtion, and also those of NPs, after protein corona coating. One first step is the investigation of HSA structural stability in the presence of Fh-NPs. Consequently, this study aims to provide valuable details about HSA structural changes induced in HSA corona by Fh-NPs. The HSA is the most frequently identified protein in the protein corona of NPs [29]. This study could be a starting point for generating bio-compatible nanomaterials with controlled surface characteristics for medical applications. Employing HSA as a standard model could allow understanding and improving the knowledge on how a particular type of iron NPs binds its protein corona. Protein–ligand docking studies have been exploited to determine the probable location of the ferrihydrite and the binding affinity in the HSA structure. This study will provide elaborate information about the nature of Fh-HSA binding phenomena and can provide significant insight into future clinical research.

## 2. Results and Discussions

### 2.1. Characterization of Ferrihydrite Nanoparticles by Scanning Electron Microscopy

The morphology of the powder particles was examined by the SEM method. Figure 1a–c show the granular appearance of the samples Fh-NPs, Co-Fh-NPs, Cu-Fh-NPs. The value of the scale bar is displayed in the photos in the lower right corner. The images were analyzed with the ImageJ program and particle size distribution determined in each case (Figure 1d–f). Samples investigation shows normal size distribution particles with the following parameters: (i) 6.4 nm mean value and 0.94 standard deviation for the Fh-NPs sample; (ii) 10.5 nm and 1.65 respectively, in the case of Co-Fh-NPs particles; and (iii) 9.5 nm and 1.47 respectively for Cu-Fh-NPs. The obtained dimensional values are consistent with earlier results [30].

### 2.2. Characterization of HSA Corona around Fh-NPs by Atomic Force Microscopy

To study the size and morphology of Fh particles coated with HSA, an AFM analysis was performed. Using the semicontact mode AFM, we succeeded in 3D visualization and characterization of individual particles and particle groups of Fh-NPs deposited on a mica substrate. Figure 2 shows the topography (Figure 2a1,b1,c1) of the samples that allows roughness profile investigation. The contrast phase (Figure 2b1–b3) images help differentiating regions of high and low surface adhesion or hardness. The profiles (Figure 2c1–c3) associated with Fh samples ((1) simple, (2) doped with Co, and (3) doped with Cu) enable estimation of the average diameter of Fh-NPs coated with HSA.

A high density of roughly spherical nanoscale particles of all three types of Fh samples, coated with HSA corona, was observed on mica surface after drying. This is consistent with other studies of HSA in the presence of other metal ions [31,32,33], where it is shown that HSA aggregated in presence of ions, without major structural rearrangements in its structure, despite a significant aggregation. Figure 2 shows that all types of Fh particles deposited on mica/tend to form agglomerates coated with a very organized corona of HSA molecules around. The average diameter of Fh-NPs coated with HSA corona was ≈150 nm for simple Fh-NPs, ≈100 nm for Co-Fh-NPs, and ≈12 nm for Cu-Fh-NPs. In a previous study [24], we showed that all three Fh-NPs have dimensions of 6–7 nm. Thus, a HSA corona around the NPs is around 144 nm for the simple Fh-NPs, 93 nm for Co-Fh-NPs and, surprisingly, 6 nm for Cu-Fh-NPs in a quite similar way with the mentioned HSA data.

The sensitivity of Cu to form a complex with the *N*-form of HSA (at pH close to physiological, as is the case in this study) may be an explanation for the different topography of the HSA arrangement around Cu-Fh-NPs. The literature shows that Cu and Co binding to albumin is done at the *N*-terminus of the protein, but at pH 7.4, no formation of a Cu-HSA complex is observed, unlike Co-HSA [33,34]. Therefore, the organization of HSA molecules (which are *N*-form) around Cu-Fh-NPs is not favored, which is reflected in the topographic representation in Figure 2a1–a3.

### 2.3. Binding of Fh to HSA by Molecular Docking

Docking simulation for Fh-HSA complex was done using eight exhaustiveness, which is a parameter that controls how comprehensive the search is for the position of the ligand in the protein site. The results obtained following molecular docking, the optimum position of the ligand in the protein with binding affinity of Δ*G* = −4.7 kcal/mol and the apparent association constant *K_a_* = 3.04 × 10^3^ M^−1^, suggest a low affinity for binding of Fh to HSA. This result was also confirmed in fluorescence [22].

The ligand, Fh, is positioned in the HSA metal binding site (Figure 3), which is surrounded by Trp 214, Tyr 30, Tyr 84, Tyr 148, and also other residues (Ala 26, Ala 26, Ala 143, Arg 98, Arg 144, Asp 107, Asp 108, Asp 89, Asn 99, Asn 111, Cys 101, Cys 90, Glu 100, Glu 86, Gln 32, Gln 33, Gln 29, His 427, His 105, Met 87, Ile 25, Phe 27, Phe 102, Pro 110, Pro 147, Lys 106, Leu 31, and Leu 103). Distance between Trp214 and ferrihydrite is 1.26 nm.

These results regarding the identification of the amino acids residues of the binding site and the determination of binding affinity help with understanding the mechanism of Fh binding to HSA. The information supports the investigation of HSA stability based on changes in the fluorescent emission of Trp. In addition, this study could have applicability in the pharmacological/medical field, as it is known that molecular docking is used in protocols to identify and validate new drugs.

### 2.4. Trp as a Spectral Sensor to Monitor HSA Thermal Stability

HSA corona is a dynamic layer (containing also, other biomolecules from plasma) adsorbed onto the surface of NPs immediately after their contact with blood plasma [30]. By binding to Fh-NPs, but also under the influence of environmental factors, such as temperature and pH, the structure of HSA corona could suffer a denaturation. However, the stability of HSA may affect the binding parameters or the pharmacokinetics/pharmacological effects of various small molecules and drugs to its site(s).

To study the stabilization of HSA structure in HSA-Fh-NPs hybrids, a very sensitive method of measuring HSA’s intrinsic fluorescence with temperature increase was applied [35]. The strong intrinsic fluorescence emission of HSA is due to the 214-Trp residue. The changes in the microenvironment of Trp-214 from HSA, caused by temperature, as well as increases (from 25 to 85 °C) in the presence of simple Fh-NPs and Fh-NPs doped with Co and Cu were monitored by fluorescence technique.

The gradual heating of HSA causes a steady decrease of the fluorescence emission of 214-Trp and a small red-shift, which is an indication for the exposure of 214-Trp residue to the solvent during temperature increase. The degree of HSA denaturation was determined by calculating the percentage of HSA denaturation, according to Equation (1) [35]:
(1)$$P=\frac{F(T)-F_{N}}{F_{D}-F_{N}}\times 100\%$$
where *P* is the fraction of the denatured protein, *F_N_* and *F_D_* are the fluorescence intensities of the native and denatured states of the protein, and *F(T)* is the fluorescence intensity at temperature, *T*.

The thermal denaturing processes of apo-HSA and HSA in the presence of simple and doped Fh-NPs are presented in Figure 4. A descendent profile of the curves was observed for the changes in HSA fluorescence intensity (*F*_max_) in the absence of Fh-NPs. This behavior was attenuated in the presence of all types of NPs, suggesting a stabilization of HSA structure in hybrids. This is in accord with the fact that the thermal stability of the proteins increases when a ligand is bound to the protein site(s) [36].

Considering HSA thermal denaturation as a reversible reaction between two states, native (*N*) and denatured (*D*), one can follow the equilibrium when the concentrations of unfolded (denatured) and folded (native) protein molecules are the same. One can calculate the percentage of the denaturation and equilibrium constant using Equation (2) [35]:
(2)$$K_{eq}=\frac{\left[D\right]}{\left[N\right]}.$$


The percentage of denatured states (Figure 5 left) is enhanced by heating temperature in the range (25–85) °C. It can be observed that up to 35 °C, the denaturation process is slow, and as the temperature increases, the HSA conformation becomes completely denatured. The equilibrium constant *K_eq_* can be calculated from the fluorescence maxima corresponding to *N* and *D* states, according to Equation (2).

The temperature of denaturation, *T*, may be determined on the basis of a second-order Van’t Hoff equation (Equation (3)) using one of the solutions of the quadratic Equation (4) [35].
(3)$$lnK_{eq}=A+BT^{-1}+CT^{-2}$$
(4)$$T=\frac{2C}{B+\sqrt{B^{2}-4AC}}$$


The graphical representation is presented in Figure 5 (right), where ln *K_eq_* is represented as a function of 1/*T*. The denaturation temperature can be estimated at around 55 °C for all HSA-NPs hybrid samples, but for the HSA-Co-Fh-NPs sample, a shift to 60 °C was observed.

The variation of Gibbs free energy, Δ*G*, of a protein, on changing from the native (*N*) state to the denatured (*D*) can be related to the equilibrium constant (Equation (5)). This parameter depends on the exposure of the internal non-polar and polar groups of protein molecule and their interaction with the solvent (i.e., water) [37]. The variation of the Gibbs free energy, Δ*G*, during denaturation (i.e., transition from *N* to *D* states), can be determined with the Equation (5) [35] and the representation of Δ*G*, as function of temperature, is illustrated in Figure 6:
(5)$$\Delta G=-RT\;ln\;K_{eq}.$$


The variation of Gibbs energy, Δ*G,* is negative, as it was expected, for all samples (Figure 6). The alteration of HSA structure, after heating above 55 °C, is attributed to the partial unfolding of HSA by its previous complexation with Fh-NPs. The denaturation temperature can be estimated around 55 °C for all HSA-NPs hybrid samples, except for the HSA-Co-Fh-NPs sample, for which the denaturation temperature is shifted by 5 °C to 60 °C (Figure 5 and Figure 6). This different behavior induced by the presence of Co^2+^ dopant may be due to the formation of chelates, thus modifying the activity of the Fh-NPs, and also its mechanism of the interaction with HSA.

These studies contribute to the understanding of how Fh-NPs characteristics (type, shape, size, surface), but also environmental conditions (such as temperature variation) induce structural changes in serum albumin, influencing its stability. Similar studies performed on BSA and Au-NPs [38] showed that Au-NPs have a similar behavior, different types of AuNPs inducing different variations in the stability, structure, and function of albumin.

## 3. Materials and Methods

Human serum albumin (purity over 98%) was purchased from SIGMA. Protein concentration was measured with a *Perkin Elmer Lambda 2S* spectrophotometer (Waltham, MA, USA.), using the standard molar absorption coefficient at 280 nm (*ε* = 36,000 M^−1^ cm^−1^) [24,39].

Ferrihydrite nanoparticles. Synthetic ferrihydrite particles, simple and doped with Co, were prepared as described earlier [40,41]. The copper-doped ferrihydrite sample was obtained using a similar procedure but with the addition of copper (III) salt to the reaction solution. The structural and magnetic properties of the samples have been recently reported [30,40,41]. Magnetization, ferromagnetic resonance, and Mossbauer spectroscopy measurements of Fh-NPs and Co-Fh-NPs samples allowed the determination of the nanoparticle size, the blocking temperature, the coercive field, the saturation magnetization, and the antiferromagnetic susceptibility [40,41]. Studies of Fh-NPs, Co-Fh-NPs, and Cu-Fh-NPs using small-angle neutron scattering and transmission electron microscopy made it possible to carry out a three-dimensional and, respectively, two-dimensional structural analysis of samples and determine the morphology and sizes of particles and their aggregates in each case [30,40,41].

Scanning Electron Microscopy (SEM). Field-emission high-resolution scanning electron microscope, type FESEM SU-8020 (Hitachi, Tokyo, Japan) was used for the morphological analysis of simple synthetic ferrihydrite particles. The view was performed in secondary electrons (SE) recording mode at acceleration voltage 10 kV. For SEM studies, the samples were prepared using the standard method. A layer of conducting glue (colloidal silver) was applied to the specimen stage, and powder particles were deposited on the glue; then, to remove the surface charge, everything was sprayed with a layer of gold–palladium alloy. For these purposes, a Q150R S magnetron sputter coater was used. The sprayed alloy layer is about 10–15 nm.

Buffer solution. The samples were prepared in 100 mM HEPES (4-(2-hydroxyethyl)-1-piperazineethanesulfonic acid) buffer (molecular weight of 238.3 g/mol) at pH 7.25 using a InoLab 720 pH-metter.

Atomic Force Microscopy (AFM). Topography and phase contrast images were obtained by means of SPM-NTegra Prima AFM (NT-MDT Spectrum Instruments, Moscow, Russia), operated in semicontact mode, using a NSG 01 cantilever (resonance frequency: 87–230 kHz, force constant: 1.45–15.10 N/m). Samples were deposited on freshly cleaved mica, air dried at room temperature, and then washed with distilled water to remove the additional material.

Molecular docking. The analyzed protein, HSA (1AO6) [42], was retrieved from Protein Data Bank [43,44]. The structure of the ferrihydrite was retrieved from Crystallography Open Database [45]. Determining the binding affinity and the best orientation of the ligand at the protein site was possible using *PyRx* (https://pyrx.sourceforge.io/) [46] with the *Auto Dock Vina* [47] algorithm. *UCSF*
*Chimera* (https://www.cgl.ucsf.edu/chimera/download.html), developed by the Resource for Biocomputing, Visualization, and Informatics from University of California, San Francisco, with support from NIH P41-GM103311 [48] was also used in docking process.

Fluorimetry. A Perkin Elmer LS55 spectrophotometer (Waltham, MA, USA.) was used to record the emission spectra of the protein (by excitation at 295 nm), with a scan speed of 500 nm/min, between 25 and 80 °C. All spectra were corrected in order to eliminate the inner *filter* effect.

## 4. Conclusions

SEM and AFM experiments revealed the dimensions of Fh-NPs and organization of HSA molecules around Fh-NPs, in a fairly regular corona, both for simple Fh-NPs and for those doped with Cu and Co.

The molecular docking studies show that ferrihydrite is binding, with a low affinity, in proximity of the metal binding site of HSA, which is located in subdomains IIA and IIIA.

Thermal denaturation of HSA alone and in the corona around Fh-NPs produces the aggregation of HSA molecules.

The HSA transition temperature is almost the same for all types of Fh-NPs from the hybrid samples with HSA, except for HSA-Co-Fh-NPs sample.

Currently, there is growing interest in the synthesis and use of Fh-NPs in therapeutic (hyperthermia and drug targeting) and diagnostic applications (nuclear magnetic resonance imaging imaging). Therefore, the biophysical effect of Fh-NPs on serum proteins is a first step in their use in biological applications.

## Figures and Tables

**Figure 1 ijms-21-09734-f001:**
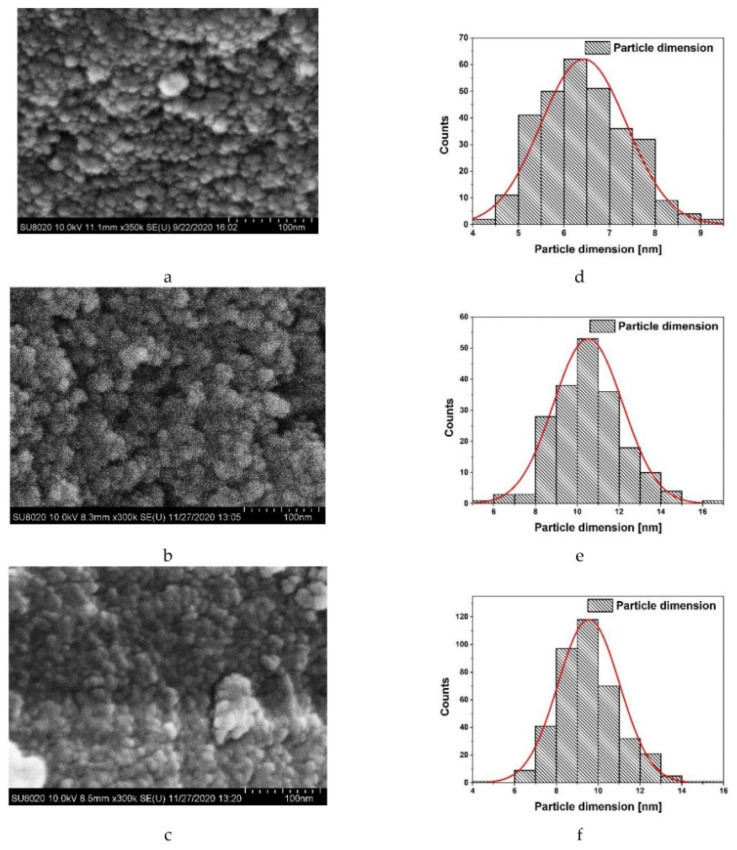
SEM image of particles obtained with a FESEM SU-8020 Hitachi instrument: ferrihydrite nanoparticles (Fh-NPs) (**a**); Co-Fh-NPs (**b**); Cu-Fh-NPs (**c**); particle size distribution hystogram and normal size distribution curve for each case (**d**), (**e**), and respectively (**f**).

**Figure 2 ijms-21-09734-f002:**
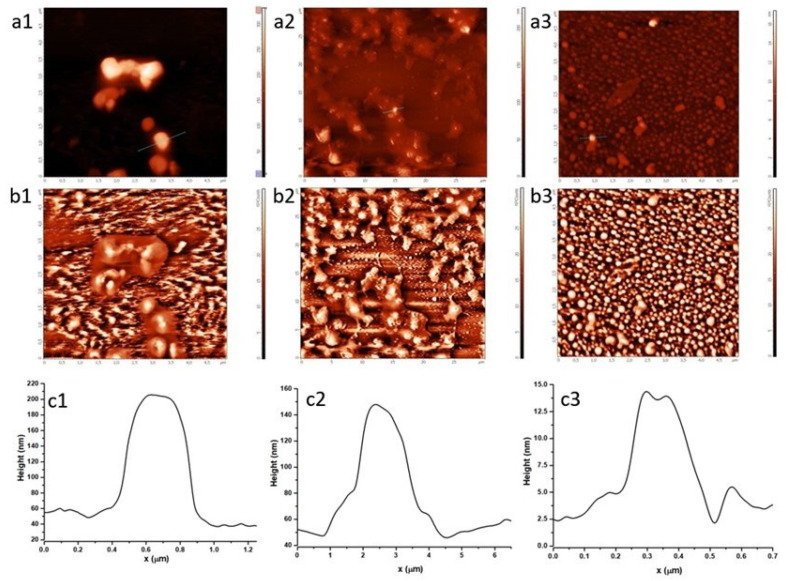
The topography (**a1**–**a****3**) and contrast phase (**b1**–**b****3**) images and the profiles (**c1**–**c****3**) of simple Fh-NPs, doped with Co, and Cu Fh-NPs coated with human serum albumin (HSA), deposited on mica substrate.

**Figure 3 ijms-21-09734-f003:**
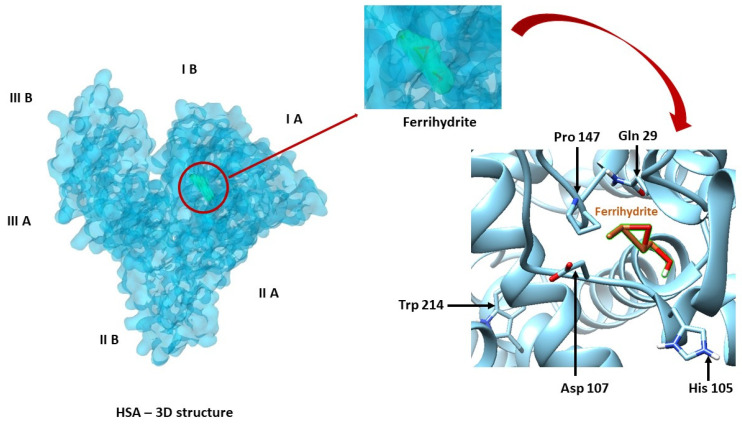
Molecular docking results for the best orientation of the ferrihydrite molecules in the binding pocket of HSA. Some catalitic residues of the binding site are represented—Gln 29, His 105, Asp 107, Pro 147, and Trp 214.

**Figure 4 ijms-21-09734-f004:**
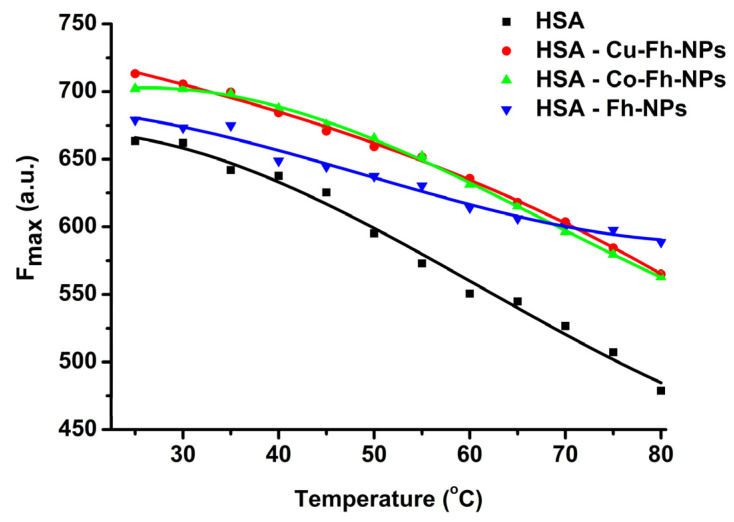
Profiles of thermal induced unfolding of HSA (black), HSA-Fh-NPs (dark cyan), HSA-Co-Fh-NPs (blue) and HSA-Cu-Fh-NPs (red), monitored by fluorescence emission (λ_ex_ = 295 nm). HSA (2.5 µM) was dissolved in 100 mM HEPES, at pH 7.25.

**Figure 5 ijms-21-09734-f005:**
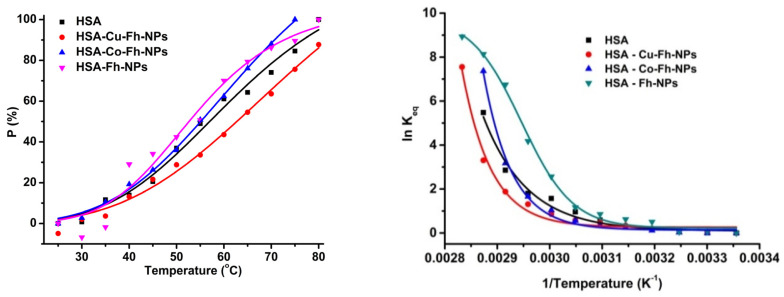
Percentage (*P*) of HSA denaturation as a function of temperature (**left**). Graphical representation for calculation of the HSA denaturation temperature (**right**). HSA (■), HSA-Fh-NPs (▼), HSA-Co-Fh-NPs (▲), and HSA-Cu-Fh-NPs (●). HSA (2.5 µM) was dissolved in 100 mM HEPES, at pH 7.25.

**Figure 6 ijms-21-09734-f006:**
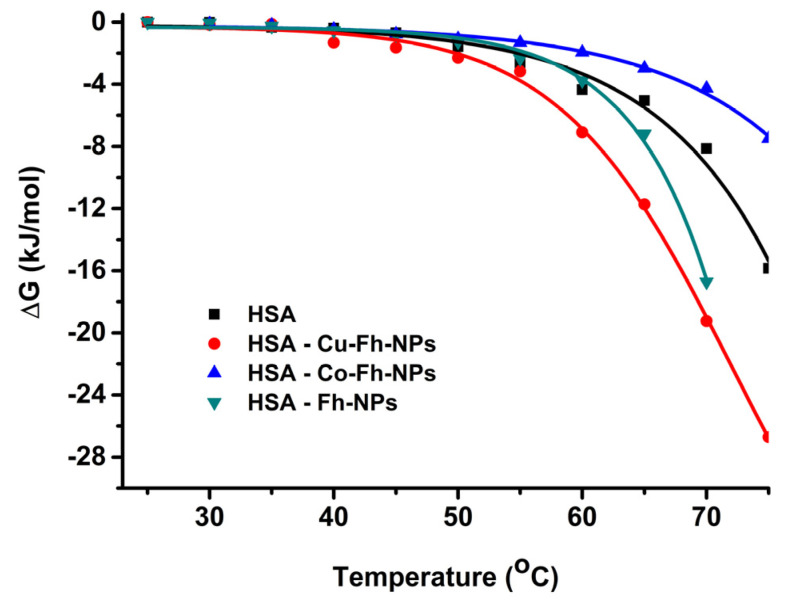
The variation of the Gibbs free energy, Δ*G,* for HSA (■), HSA-Fh-NPs (▼), HSA-Co-Fh-NPs (▲), and HSA-Cu-Fh-NPs (●), in 100 mM HEPES, at pH 7.25.

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
