# Peer review of "Exploring the Conformation and Thermal Stability of Human Serum Albumin Corona of Ferrihydrite Nanoparticles"

_ijms, 2020, doi:10.3390/ijms21249734_

Round 1

Reviewer 1 Report

The manuscript “Exploring the conformation and thermal stability of human serum albumin corona of ferrihydrite nanoparticles” contains important aspects of current interest. I recommend this paper for publication after minor revision.

Comments to Author:

There are no references in several places. Please complete the citation:

Line 152-153 “the percentage of the denaturation and equilibrium constant using equation”

Line 166-167 “This parameter depends on the exposure of the internal non-polar and polar groups of protein molecule and their ...”

Line 172 – the equation DG = ….

Line 192-193  - “the standard molar absorption coefficients of HSA”

I recommend comparing the research results with other nanoparticles interacting with proteins, eg AuNps …

Author Response

Thank you for your suggestions and observations!

Sincerely,

Claudia Chilom

Reviewer 2 Report

The manuscript „Exploring the conformation and thermal stability of  human serum albumin corona of ferrihydrite  nanoparticles” describes the corona formation in size, binding affinity and stability of human serum albumin around ferrihydrite nanoparticles.

The manuscript is well written and discussed. The citations are balanced and the experiments are described soundly.

I really like the study. However, I would like to raise a few issues:

In general: Reference 35 only describes cobalt doped ferrihydrite nanoparticles. Copper doped particles are not described. Since you have all the characterization information on your nanoparticles, may you add them in a supporting information. I think it would be nice to have SEM/TEM images of the respective materials as well as a characterization of the nanomaterials physical properties such as Mössbauer spectroscopy.

Can you further discuss the large discrepancy in corona size obtained from AFM investigations for the copper doped and the other ferriyhdrite nanoparticles?

Furthermore, I do not know what a fystogram is. I think you mean histogram in the figure caption of figure 1.

Author Response

(The authors gave the same response as above.)

Reviewer 3 Report

This work of Chilom et al. analyzes the shape and stability of the human serum albumin (HSA) adsorbed to ferrihydrite nanoparticles. The particles shape was obtained with Scanning Electron Microscopy (SEM) and Atomic Force Microscopy (AFM), whereas the stability was measured by means of thermal unfolding followed by intrinsic fluorescence of a Trp residue present in HSA. Molecular docking was also performed to find the position of the ferrihydrate in the HSA structure. While the interest of ferrihydrate particles and their interaction with a protein such as HSA is high, the conducted research and its presentation in its current form does not deserve publication in a prestigious journal as IJMS. The main flaws according to this referee are:

- SEM and AFM are powerful techniques; however, herein no controls are depicted, the writing is non self-explanatory, and the significance of the obtained figures is missing.

- Similarly happens to molecular docking. Experiments are made, but the consequences of results are not addressed.

- Regarding the thermal stability, research has not been properly analyzed; therefore, conclusions are not in accord to the experimental results.

- English writing is poor, and several errata are found along the paper.

Specifically, to be more concrete in some of the previous comments:

  1. In Figure 1, a fystogram is mentioned, but there is no explanation about what it is and how it is generated.
  2. In Figure 2, three kinds of pictures are shown (topography, contrast phase and profiles); however, they are not explained and why there are three kinds.
  3. It is stated that docking simulation “was done using 8 exhaustiveness” (line 105). One cannot know what is this sentence referring to. Also, in the following paragraph (starting in line 109), multiple residues are mentioned, which is uninformative. Nothing is commented nor indicated in Figure 3, showing the relevance or interest of those residues.
  4. The fluorescence curves in the thermal unfolding are poorly sigmoidal, so an analysis is difficult to made. Even more to assert that it is reversible and that follows a two-state model (line 150). Indeed, in line 159 it is said that “intermediate structures” are found, which is contradictory to the two-state model. Reversibility has to be checked, which has not been done. Finally in this section, Tm has been determined by a representation of ln Keq as function of 1/T, but how is the value obtained?
  5. In the first paragraph of the conclusions, it is said that the arrangements are a “fairly regular corona”, but this has not been described previously. In the third paragraph, again intermediates in the thermal unfolding are stated, together with aggregation. Aggregation processes are usually linked to irreversibility in thermal unfolding. And the last paragraph of the conclusions reads that the biophysical knowledge is useful for potential medical applications, although nothing about reasons or ways of this usefulness is said.
  6. English writing mistakes are found along the manuscript (examples: line 43 “corona proteins can reduces…”; line 70 “Employing…”; line 59 “programme”; etc.). Also some sentences are poorly or improperly constructed, e.g., lines 93-94 “to form agglomerates coated with a very organized HSA around”, line 133 “The degree of HSA denaturation was followed calculating the percentage”, lines 158-159 “but as the temperature increases, more and more, the HSA molecules…”.

Author Response

(The authors gave the same response as above.)

Round 2

Reviewer 3 Report

This reviewer congratulates the authors, since the manuscript has significantly improved upon their introduced changes. Most of my concerns have been successfully addressed. Nevertheless, one significant issue has not yet been correctly solved. I can admit the reasons given by the authors about reversibility in the thermal denaturation of HSA. But in line 181 of the revised manuscript, the authors state that HSA molecules are in intermediate structures, whereas they keep saying that the thermal denaturation follows a two-state regime (line 172). The authors have to see that both statements are a contradictio in terminis, i.e., they contradict each other. Or it is a two state model or it has intermediate structures. I am fine with either solution they find best, but they cannot say both things at the same time. 

Furthermore other details to be improved can be found:

  • In the response letter, the authors state that they clarify the meaning of exhaustiveness, whereas in the revised manuscript is missing.
  • The way of calculating Tm is still missing (no explanation is found, either in the response letter or the manuscript).
  • English writing has been improved, but there are still changes to be made. 

Author Response

We are grateful for your comments. We consider that, responding to your requirements, the quality of our work is better. Regarding the English language, we will use the services offered by MDPI, to avoid any incorrect expressions. We hope you agree with the publication of the article in this latest version.

Round 3

Reviewer 3 Report

Now I am totally fine with the publication of the manuscript in IJMS in its present form. Regarding my opinion now, congratulations!!